# Building Agro-Industrial Capabilities in the Sugarcane Supply Chain in Brazil

**Gabriel da Silva Medina [1],\* and Rommel Bernardes da Costa [2]**

[1] Faculty of Agronomy and Veterinary Medicine, University of Brasília, Brasília 70910-900, Brazil
[2] Faculty of Agronomy, Federal University of Goias, Goiânia 74690-900, Brazil; rommelbc@ufg.br
\* Correspondence: gabriel.medina@unb.br

**Abstract:** *Background:* This study aims to explore how domestic entrepreneurs can benefit from the thriving global agribusiness by establishing themselves in agro-industrial segments that can best remunerate capital and labour. The ways in which domestic entrepreneurs in Brazil enter different segments of the agribusiness industry were assessed with specific attention to implications for the development of local agro-industrial capabilities. *Methods:* We assessed the current market share of domestic companies in relation to foreign multinationals in various segments of the sugar and ethanol supply chain in Brazil. *Results:* Foreign multinationals are market leaders in the fertilizers, machinery and trading segments (domestic companies market share is 20.3%, 33.3% and 42.9% in those segments respectively). However, Brazilian companies have achieved higher market share in segments such as plant breeding, sugarcane processing and farming (domestic market share is 93.2%, 67.4% and 75.5% respectively). Plant breeding, farming and trading benefit from governmental support in research, subsidized credits and market policies respectively. *Conclusions:* By investing in agro-industrial sectors developing countries can benefit from agribusiness expansion for their economic growth. Investments in science and technology and domestic regulatory actions can help to build country capabilities, although the impacts are sometimes limited to the agro-industrial sectors where domestic companies are more competitive. These lessons can help other developing countries to assess their opportunities and challenges for agro-industrial development.

**Keywords:** development economics; foreign direct investments; sugar-energy sector; food chain management; logistics and the competitiveness of firms and places





## 1. Introduction

Agribusiness is one of the most dynamic economic sectors in some developing countries such as Brazil, leading to debates on how to build country agro-industrial capabilities that go beyond farming [1,2]. With the relative loss of industrial share in the economy, agribusiness has become one of the main drivers of the Brazilian economic growth and is particularly important for the country's positive trade balance [3,4]. In 2022, agricultural production responded for 7% of the Brazilian Gross Domestic Product (GDP), whereas agribusiness as a whole (including inputs, agricultural production and industrial processing) responded for 24.8% of the national GDP [5].

Classical economics and the theory of associated dependent development state that developing nations such as Brazil can grow by associating themselves with developed nations as a means to attract private foreign direct investments (FDI) [6,7]. In contrast, neoclassical economics and the new developmentalism theory suggest that state-led structural change towards a more sophisticated industrial base is a sine qua non condition for an emerging economy to converge with developed ones [3,8].

The liberal and globalized business environment in which Brazil is now embedded requires a new development paradigm based on opportunities created by dynamic economic sectors such as agribusiness [9]. A crucial challenge is the consolidation of domestic

companies along the agribusiness supply chains established in Brazil. Studies reveal that investments in agro-industrial sectors have greater positive effects than those investments made in other economic sectors [10].

The trade liberalization that took place in Brazil during the 1990s led to large investments in Brazilian agribusiness, mainly by foreign multinational corporations [11]. However, foreign investments did not occur in the same way in all supply chains. For example, while the soybeans supply chain now has a predominance of multinational groups in their agro-industrial sectors [12,13], the sugar and ethanol supply chain has more domestic investments, including ventures in the technological and industrial sectors [14].

The sugar-alcohol supply chain had a great development in Brazil from 1973 onwards with the creation of the National Alcohol Program [15]. In the late nineties, the sugar and ethanol industry underwent a process of deregulation. From the 2000s onwards, with the advent of flex-fuel vehicles, the supply chain was modernized but continues to rely on state support for the differential federal taxes between gasoline and ethanol, for the mandatory mixture of 27% of ethanol in gasoline and for the Renovabio program to increase ethanol consumption, in addition to tax benefits in different states [16].

As a result of foreign investments and domestic policies, since 2012 the sugar and ethanol supply chain, including the segments of inputs, farming and industry, accounts for 10% of the GDP of the Brazilian agribusiness [17]. Brazil is the world's largest producer of sugarcane, with 654.7 million tons processed in the 2020/21 harvest [18], and the second largest global producer and exporter of ethanol and sugar [17]. With economic liberalization, waves of mergers and acquisitions were unleashed in Brazil resulting in four publicly traded leading companies Cosan, São Martinho, Tereos and Petrobras [19].

This study aims to explore how domestic entrepreneurs can benefit from the thriving global agribusiness by establishing themselves in agro-industrial segments that can best remunerate capital and labour. We seek to reveal how domestic entrepreneurs in Brazil enter different segments of the agribusiness industry, with specific attention to implications for the development of local capabilities. For such, we assess the participation of Brazilian capital in the sugar and ethanol supply chain established in Brazil as a means to identify opportunities for strategic domestic investments in agribusiness. Specifically, this study is intended to:

- Estimate the market share of Brazilian groups in relation to foreign multinationals in the main segments of the sugar and ethanol supply chain;
- Identify strategic areas for policies to support domestic agribusiness based on segments with the potential for greater participation by local groups.

Building on that information, we intend to explore the possibilities for a new development paradigm based on opportunities created by dynamic economic sectors in developing countries that can build country capabilities and resilience in agro-industrial segments [20]. By investing in the agro-industrial sectors that better remunerate capital and labour, and going beyond the current focus on the primary production of commodities, developing countries can benefit from agribusiness expansion for their development [13,21].

## 2. Methods

The starting point of this study was the identification of the most commonly used inputs for each stage of the sugar and ethanol supply chain, its suppliers and the country of origin of the shareholders of the companies that supply these inputs. This was done by reviewing the specialized literature and consulting the institutional material of the companies and their sectoral associations. As a result, the following main segments of the Brazilian sugar and ethanol supply chain were identified: plant breeding, fertilizers, machinery, mills (processing plants), farming and trading.

Business associations organized by sector, such as the Sugarcane Industry Union (Única), estimate the participation of their members in the market and publish this information in statistical yearbooks that are available on their internet pages. Based on information on the participation of companies in each segment made available by the

business associations, the participation (market share) of companies with domestic capital was estimated. The total quantity produced in each segment was identified and then the productive capacity of the main companies operating in the segment was surveyed. Sources are summarized in Table 1 and also cited throughout the work.

**Table 1.** Business associations' websites used as sources of information.

| Segments | Organizations | Weblinks | Reference * |
|---|---|---|---|
| Plant breeding | Interuniversity Network for the Development of the Sugarcane Sector (RIDESA) | https://www.ridesa.com.br/ (accessed on 20 December 2022) | Ridesa, 2022 [22] |
| Fertilizers | Brazilian National Fertilizer Association (ANDA) | http://anda.org.br/ (accessed on 20 December 2022) | ANDA [23] |
| Machinery | National Association of Motor Vehicle Manufacturers (ANFAVEA) | http://www.anfavea.com.br (accessed on 20 December 2022) | Anfavea, 2022 [24] |
| Mills | Brazilian Sugarcane Industry Association (UNICA) | https://unica.com.br/ (accessed on 20 December 2022) | Unica, 2022 [17] |
| Farming | Organization of Associations of Sugarcane Producers in Brazil (Orplana) | https://www.orplana.com.br/ (accessed on 20 December 2022) | Orplana [25] |
| Trading | NovaCana | https://www.novacana.com/ (accessed on 20 December 2022) | NovaCana [26] |

* The specific sources used for the different analyses are mentioned in the results section.

The shareholding structure of publicly traded companies is protected by law in Brazil; as a result, multinational companies are not obliged to make this information publicly available. Thus, it was often challenging to determine whether a company was controlled by a Brazilian or multinational company because multinationals tend not to share this information. A further difficulty arose from the fact that multinational corporations tend to sell their products with the brand names of the Brazilian companies that they acquired [27].

However, all publicly traded companies disclose the shareholding composition of the controlling groups on their websites. Thus, it was possible to identify whether the companies were controlled by Brazilian or foreign groups. In the case of companies headquartered in the State of São Paulo (which accounts for 55% of the sugarcane planted area in the country), information on partners or directors and corporate purposes is made public by the Board of Trade.

The market shares of all companies with Brazilian capital were summed ($\sum_{i=1}^{n} Br_i$) to estimate the total market share of domestic groups (D) in each segment of the supply chain. Domestic share in the supply chain (DS) resulted from the sum of the participation of business groups with Brazilian capital in each of the six segments analyzed (from the improvement of sugarcane varieties to the commercialization of sugar and ethanol) (see Equation (1)). The sum of the domestic share is presented in the results section.

$$DS = (D1 + D2 + D3 + D4 + D5 + D6)/6, \text{ where } D = \sum_{i=1}^{n} Br_i \tag{1}$$

where: DS—Domestic participation in the supply chain; D—Domestic participation in each segment (from slaughterhouses to seeds); $\sum_{i=1}^{n} Br_i$—Sum of the domestic participation of each segment of the supply chain.

## 3. Results

Figure 1 presents the main segments of the sugar and ethanol supply chain and also summarizes the results of this study on the domestic market share for each segment. The results for each segment are presented in detail in the results subsections.

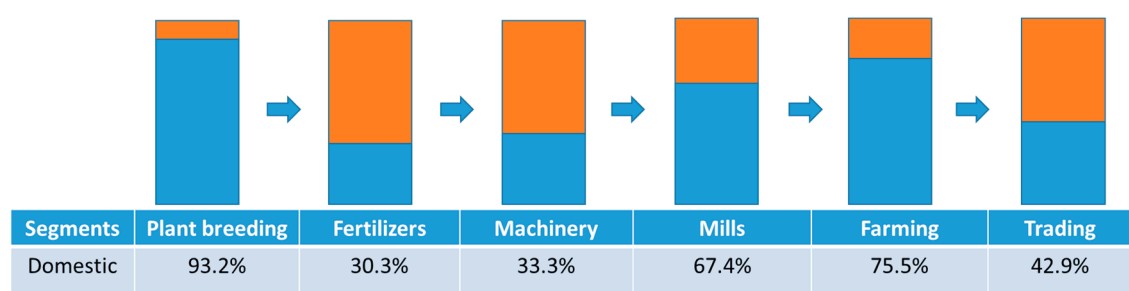

Legend

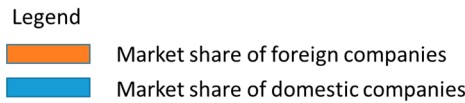

Market share of foreign companies

Market share of domestic companies

**Figure 1.** Key segments of the sugar and ethanol supply chain in Brazil.

### 3.1. Plant Breeding

Sugarcane plant breeding in Brazil is, to a large extent, done by domestic companies that count on support from public agencies for research and development (R&D). Sugarcane varieties developed by the Interuniversity Network for the Development of the Sugar-Energy Sector (Ridesa), named with the acronym RB, are cultivated in more than 65% of the sugarcane area in Brazil [22]. The other leading varieties are CTC (with 14% of the planted area), SP (with 13% of the planted area), IAC (with 2% of the planted area), CV (with 2% of the planted area) and others (with 4% of the planted area) (Table 2). Ridesa has 10 universities, 79 research bases and different selection areas, the latter conducted in partnership with private companies that pay royalties to adopt the innovation developed by Ridesa. As the royalty paid to Ridesa is often lower than the amount paid to private plant breeders, Ridesa varieties tend to hold the largest share of the market.

**Table 2.** Market share of leading companies in the sugarcane variety development segment for planting in 2021.

| Varieties | Developer | Headquarters | Market Share (%) | Domestic Share (%) |
|---|---|---|---|---|
| RB | RB—Ridesa (network of universities) | Brazil | 65 | 65 |
| CTC | Copersucar, Raízen and BNDESPar | Brazil (80%) | 14 | 11.2 |
| SP | Copersucar | Brazil | 13 | 13 |
| IAC | Instituto Agronômico de Campinas (São Paulo government) | Brazil | 2 | 2 |
| CV | CanaVialis (Monsanto/Bayer) | Germany | 2 | 0 |
| Others | | | 4 | 2 |
| Total | | | 100 | 93.2 |

Source: Ridesa, 2022 [22].

The Sugarcane Technology Center (breeder of the CTC varieties) was created in 1969 by the group of Copersucar mills with a focus on conventional plant breeding and, recently, directed investments towards sugarcane transgenics. Since 2017, the company has already launched two transgenic varieties resistant to the sugarcane borer. The second wave of transgenics foresees the release of a weevil-resistant variety. The company expects to reach the end of the 2022/23 harvest with 50,000 hectares planted with transgenic sugarcane, out of a total of 8.13 million hectares planted with sugarcane across Brazil. Among CTC's main shareholders are Copersucar, with 26% of the business, Raízen (a joint venture between Cosan and Shell), with a 20% share, and the equity branch of the National Bank for Economic and Social Development (BNDESPar), with 18.9%.

The only CTC competitor in the transgenic segment in Brazil is CanaVialis (breeder of the CV varieties). CanaVialis was created in 2003 by Brazilian scientists who took part in the sequencing of the sugarcane genome and had the Votorantim group as its main

investor. The business caught the attention of Monsanto, which bought the company in 2008. Seven years later, the multinational decided to leave the sugarcane sector and ended the company's activities in Brazil. As the CV varieties had a patent in the National Registry of Seeds and Seedlings (Renasen), Bayer (current owner of Monsanto) continues to receive royalties for the use of its varieties. Varieties of Brazilian groups still include SP from Copersucar and IAC developed by Instituto Agronomia de Campinas. An important Brazilian group that is no longer operational is Vignis, which developed VG varieties. Created in 2010, Vignis conducted the largest Cana Energia genetic improvement program in the world. In 2018, the Vignis Group had its request for judicial recovery accepted and left the market.

### 3.2. Fertilizers

Two types of companies operate in the fertilizer segment: those that produce raw materials and intermediate products (or simple fertilizers) and those that manufacture formulated fertilizers. Most of the raw material for fertilizers used in Brazil is imported by multinational companies. In the case of macronutrient phosphorus, potassium and nitrogen, respectively 44%, 95% and 75% of the total amount consumed in the country are imported [13]. In Brazil, it is common for part of the fertilization of sugarcane plantations to be done with by-products from the sugarcane plants, such as vinasse and filter cake, which reduces the cost of fertilizers.

Fertilizer manufacturing in Brazil has strong participation of the multinational Yara, with national groups holding 29.8% of the market in 2021 (Table 3). In 2018, the Canadian company Nutrien was created from the merger between Agrium and Potash and already appears to hold 10% of the Brazilian market. Following the example of multinationals in the agricultural trade, such as Archer Daniels Midland, Bunge, and Cargill, Louis Dreyfus reduced its investments in fertilizers due to poor growth prospects. At the same time, major players in the fertilizer industry (such as Yara) are buying up their competitors.

**Table 3.** Market share of companies manufacturing fertilizers used for soy in Brazil in 2021.

| | Companies | Headquarters | Market Share (%) | Domestic Share (%) |
|---|---|---|---|---|
| Phosphorous | Vale (now Mosaic) | Brazil (now US) | 53 | 0 |
| | Anglo American | UK | 12 | 0 |
| | Others | Brazil/Multinacionals | 35 | 17.5 |
| | Total (56% of domestic production) | | | 9.8 |
| Potassium | Vale (sold to Mosaic) | US | 100 | 0 |
| | Total (considering 5% of that used in the country of national production) | | 8 | 0 |
| Nitrogen | Proquigel Química S.A | Brazil | 15 | 15 |
| | Subtotal (15% of national production) | | | 15 |
| | Subtotal (Average production of nitrogen, phosphorus and potassium) | | | 10.8 |
| Manufacture of formulated fertilizers | Yara | Norway | 25 | 0 |
| | Mosaic/ADM | USA | 20 | 0 |
| | Dreyfus | France | 0 | 0 |
| | Nutrien | Canada | 10 | 0 |
| | Grupo Fertipar | Brazil | 15 | 15 |
| | Heringer | Brazil (56% national) | 6 | 3.3 |
| | Regional companies | Brazil | 7 | 6.5 |
| | Others | Brazil/Multinationals | 17 | 5 |
| | Subtotal | | | 29.8 |
| Total (%) (Average Brazilian participation in the production of raw materials and fertilizers) | | | | 20.3 |

Source: ANDA, 2022 [23].

The Fertipar Group and Heringer are the Brazilian companies with the greatest participation in the manufacture of fertilizers in Brazil. The two companies produce basic fertilizers, NPK formulations and speciality fertilizers. Facing financial hardship for some years, Heringer (a publicly traded company with 56% domestic capital) filed for receivership in 2019 and lost an important part of the market with the closure of part of its factories. The rest of the Brazilian fertilizer market is served by regional domestic companies, such as Adubos Araguaia and Fertilizantes Tocantins.

### 3.3. Machinery

The segment of agricultural machinery used for sugarcane production mainly includes planters and seeders, sprayers, harvesters and transfers. In the case of planters and seeders, there is an important participation of Brazilian groups, with emphasis on the companies DMB Máquinas e Implementos Agrícolas Ltd., TMA Máquinas (from the Tracan Group) and Sollus Agrícola. Among multinationals, American John Deere has the largest market share. Mechanical planting represents about 52% of the planted area in the traditional sugarcane area in São Paulo and Paraná, compared to 48% of manual planting, mainly due to technical problems with the machines that still need to be solved by the industry as a way to optimize the distribution of stalks, reducing planting failures.

The market for sprayers and other implements is led by the Brazilian Jacto. The French Berthoud and the multinational giants Valtra, Case and John Deere still operate in the sprayer market. In the manufacture of other agricultural implements, the Brazilian companies Civemasa also stand out in the manufacture of a wide range of implements and Teston in the manufacture of transfers.

In the case of harvesters, the market is completely controlled by the three giant multinationals in the sector (Table 4). CNH leads domestic sales of sugarcane harvesters with 348 units sold in Brazil in 2021, closely followed by John Deere with 290 units. AGCO traditionally has a smaller share of this market, having sold seven sugarcane harvesters in 2021 [24]. Santal Equipamentos S.A. was one of the Brazilian pioneers in the mechanization of sugarcane harvesting but was sold to the transnational AGCO in 2012. In 2020, the Brazilian Jacto launched the Hover 500 cane harvester, but sales data are not yet available.

**Table 4.** Market share of leading companies in the manufacture of sugarcane planters and harvesters sold in Brazil in 2021.

|  | Companies | Headquarters | Market Share (%) | Domestic Share (%) |
|---|---|---|---|---|
| Agricultural machines and implements | DMB, TMA, Sollus, and others | Brazil | 50 | 50 |
|  | John Deere and others | Multinational | 50 | 0 |
| Harvester | AGCO (Valtra) | US | 1.1 | 0 |
|  | CNH (Case) | Italy | 54.0 | 0 |
|  | John Deere | US | 45.0 | 0 |
| Industrial equipment | Dedini, Zanini Renk and others | Brazil/Multinational | 100 | 50 |
| Total (average) |  |  |  | 33.3 |

Source: Anfavea, 2022 [24].

In addition to agricultural machinery, there is the industrial equipment segment. The base industry has generalist equipment for sugarcane milling and specific equipment for the production of sugar, ethanol and energy. The main industrial equipment common to all processes is the mill or diffuser. Through mills or diffusers, the raw material responsible for the production of sugar, ethanol and bioenergy is removed from sugarcane.

Brazilian groups hold the largest share of the basic equipment market for the sugar-energy industry. However, Brazilian companies operating in this segment generally establish partnerships or joint ventures with multinational groups for the development or

importation of technologies. For example, Dedini S/A Indústrias de Base is a company with a family structure and Brazilian capital that supplies equipment from the reception, preparation, extraction and treatment of the broth to the production of sugar and ethanol. With financial difficulties since the end of 2008, in 2019 Dedini signed a technology agreement with the Indian PRAJ Industries that will manage technology licensing and supply key equipment. Zanini Renk was born as a joint venture between the companies Zanini and Renk AG, from Germany. Since 1983, Renk AG has maintained a technology transfer contract with Zanini Renk, ensuring the same technological conception developed in the gearboxes designed in Germany. The Brazilian company Sermasa Equipamentos Industriais Ltd.a. started its activities in 2006 and established important partnerships with multinational companies that own technologies and also supply complementary components of the systems, enabling it to manufacture and assemble complete plants for the sugar and ethanol sector. Founded in 1989, the Brazilian RG Sertal stands out among suppliers of services and products for the sugar and ethanol industry.

*3.4. Sugarcane Processing (Industrial Plants/Mills)*

Brazil has 234 plants that produce both ethanol and sugar and 178 distilleries that produce ethanol, summing up to 412 agro-industrial units (mills). In addition to ethanol and sugar, all industries in the sugar-energy sector produce energy and around 30% of the agro-industrial units also co-generate energy from sugarcane bagasse, selling the surplus electricity produced. The industries of the Center-South region concentrate more than 90% of the national production of 642.7 million tons processed annually [18].

The Copersucar S/A group processed 85 million tons of sugarcane in 2019, the highest volume in the country. The Company has exclusivity in the sale of sugar and ethanol produced by 34 partner mills belonging to 20 different economic groups (Table 5). The group's largest producer is Zilor, which processes 10.6 million tons of sugarcane annually, followed by Viracool with 10 million tons per year. Copersucar and its partner mills are autonomous companies and conduct their corporate policies independently.

The second largest sugar mill group in the country is Raízen, a 50:50 joint venture between Cosan S/A and Royal Dutch Shell. The partnership provides for the production of ethanol by Cosan's plants and distribution at Shell service stations. Raízen annually processes 73 million tons of sugarcane and, considered individually, is the largest segment company operating in Brazil [28]. Although known for producing ethanol, Raízen is also a major sugar producer. In 2019, the joint venture acquired NovAmérica's agricultural operations at the Caarapó (MS) unit, expanding its production capacity.

Among essentially Brazilian companies, the São Martinho Group leads the ranking of sugarcane processing and profitability in the activity. The São Martinho Group led the ranking Value in net income with a positive R$ 288.3 million in the 2014 fiscal year. In second place came Usina Colombo and, in third place, was the Santa Terezinha Participações Group. Usina Ipiranga ranked fourth. Following, they complete the list of 10 sugar-energy groups with positive net profit: Usina da Pedra; Batatais plant; Balbo Group; Adecoagro Brazil; Usina São João and the Zilor Group [29].

Characteristically, in addition to public credits, the largest Brazilian groups constituted as S/A go public and raise funds for investments either via debentures or the sale of shares. As an example, Ipiranga Agroindustrial S/A raised BRL 200 million in 2019 with debentures. Therefore, it is likely that many of these companies have creditors in other countries, although they remain companies with the national capital.

However, the segment has also attracted multinational groups. Bunge, which already operated in the market with plants and distilleries, 2019 set up a 50:50 joint venture with BP British Petroleum that resulted in the creation of BP Bunge Bioenergia. The joint venture has a total of 11 mills with 32 million tonnes of combined crushing capacity per year. In addition to sugar and ethanol, the company also co-generates electricity.

**Table 5.** Market share of the leading sugarcane processing plants and distilleries in 2021.

| Company | Headquarters | Capacity (Millions of Tons) | Market Share (%) | Domestic Share (%) |
|---|---|---|---|---|
| Copersucar | Brazil | 85 | 13.2 | 13.2 |
| 1. Zilor | Brazil | 10.6 | 1.6 | 1.6 |
| 2. Viralcool | Brazil | 10 | 1.6 | 1.6 |
| 3. Cocal | Brazil | 8.7 | 1.4 | 1.4 |
| 4. Furlan | Brazil | 6.9 | 1.1 | 1.1 |
| 5. Ipiranga | Brazil | 6.3 | 1.0 | 1.0 |
| 6. Santa Adélia | Brazil | 5.7 | 0.9 | 0.9 |
| 7. Pedra Agroindustrial | Brazil | 5 | 0.8 | 0.8 |
| 8. Melhoramentos | Brazil | 4.5 | 0.7 | 0.7 |
| 9. São Manoel | Brazil | 4.1 | 0.6 | 0.6 |
| 10. Cerradão | Brazil | 3.1 | 0.5 | 0.5 |
| 11. Ferrari | Brazil | 3 | 0.5 | 0.5 |
| 12. Umoe | Brazil | 2.8 | 0.4 | 0.4 |
| 13. Estiva | Brazil | 2.7 | 0.4 | 0.4 |
| 14. Jacarezinho (Grupo Maringá) | Brazil | 2.5 | 0.4 | 0.4 |
| 15. Pitangueiras | Brazil | 2.4 | 0.4 | 0.4 |
| 16. Uberaba | Brazil | 2.1 | 0.3 | 0.3 |
| 17. Usina São Francisco | Brazil | 1.5 | 0.2 | 0.2 |
| 18. Santa Lúcia | Brazil | 1.4 | 0.2 | 0.2 |
| 19. Caçu | Brazil | 1.2 | 0.2 | 0.2 |
| 20. São Luiz | Brazil | 0.5 | 0.1 | 0.1 |
| Raízen (Cosan/Shell) | Brazil/The Netherlands | 73 | 11.4 | 5.6 |
| BP Bunge Bioenergia | UK | 32 | 5.0 | 0.0 |
| Atvos | USA | 27 | 4.2 | 0.0 |
| São Martinho | Brazil | 24 | 3.7 | 3.7 |
| Tereos | France | 18.8 | 2.9 | 0.0 |
| Shree Renuka Sugars Ltd. | India | 13.6 | 2.1 | 0.0 |
| Usina Colombo | Brazil | 8.9 | 1.4 | 1.4 |
| Grupo Santa Terezinha Participações | Brazil | 9.1 | 1.4 | 1.4 |
| Others | Brazil | 270 | 42.0 | 42.0 |
| Others | Multinationals | 81.3 | 12.6 | 0.0 |
| Total | | 642.7 | | 67.4 |

Source: Unica, 2022 and websites of companies checked in 2022 [17].

The Atvos Agroindustrial group, which grinds 27 million tons per year, is changing from Brazilian controllers to US ones. The American fund Lone Star may be the new controller of Atvos Agroindustrial, the sugar and ethanol company of the Odebrecht group that filed for bankruptcy in 2019.

Tereos Açúcar & Energia Brasil, one of the leaders in the Brazilian sugar-energy sector, processed 18.8 million tons of sugarcane in its seven units during the 2019/2020 harvest, an increase of 7.5% compared to the previous harvest. Tereos Brasil is part of the Tereos Internacional Group, a global company of French origin that transforms sugarcane, cereals and tubers into sugar, starch, ethanol and alcohol. The plants of the Tereos Brasil group are Andrade, Cruz Alta, São José, Severínia, Mandu, Tanabi and Usina Vertente (in the latter with 50% control in partnership with the Humus Group).

Among the multinationals, the Indian group Shree Renuka Sugars Ltd. completed 2010 the acquisition of 59.4% of Equipav Açúcar e Álcool, creating Renuka do Brasil S/A with the capacity to process 10.5 million tons/year of sugarcane. Also in 2010, the Indian group acquired 100% of Renuka Vale do Ivaí with the capacity to process 3.1 million tons/per year. As a result, the Indian multinational now can process 13.6 million tons per year. Another multinational in the segment is Glencore which, through Glencane, can process 4.9 million tons annually. In 2018, Cargill acquired 100% control of Cevasa, a sugar and ethanol unit with a capacity to grind 2.3 million tons per year that it had in partnership with Canagril, although it intends to invest more in the production of ethanol

from corn, with SJC Bioenergy. In 2016, Archer-Daniels-Midland (ADM) sold the Cabreira Energética plant, its sole sugar-energy operation in Brazil. The unit came to be controlled by Companhia Mineira de Açúcar e Álcool. The plant has a crushing capacity of 1.5 million tons of sugarcane per harvest, in addition to being able to produce up to 1.2 million litres of ethanol daily.

### 3.5. Farming

It is estimated that 75% of the sugarcane farming production is carried out by the mills themselves while 25% is sourced from independent farmers in the Center-South region, which is the largest producer in Brazil. Out of the 25% market share held by independent farmers, the Organization of Associations of Sugarcane Producers in Brazil (Orplana) represents half of the production. Brazilian participation in sugarcane production was estimated at 75.5% of the total, considering that 67.4% of the industrial plant sector is controlled by Brazilian groups (Table 6).

**Table 6.** Market share of sugarcane producers for industrial purposes in 2021.

|  | Market Share (%) | Domestic Share (%) |
|---|---|---|
| Independent farmers | 25 | 25 |
| Production by the industrial group | 75 (67.4% Brazilian) | 50.5 |
| Total (average) |  | 75.5 |

Source: Orplana, 2022 [25].

The State of São Paulo, home to 26% of the total sugarcane production in Brazil, has the highest concentration of independent farmers, with sugarcane production carried out in 11 thousand farms [30]. Sugarcane farmers in São Paulo are organized around the Council of Sugarcane, Sugar and Ethanol Producers of the State of São Paulo (Consecana). Farmers from other states have already sought to create their council, but still without conclusion.

### 3.6. Trading

Ethanol and sugar trading groups were created to increase the bargaining power of producers vis-à-vis distributors [31]. The main Brazilian trading groups are Coopersucar, Bioagência, CPA and SCA (Table 7).

**Table 7.** Market share of market-leading ethanol and sugar trading companies in 2021.

| Ethanol | Headquarters | Billions of L | Market Share (%) | Domestic Share (%) |
|---|---|---|---|---|
| Raízen | Multinational | 16.5 | 49.1 | 24.5 |
| Copersucar/Eco-Energy | Brazil | 4.8 | 14.3 | 14.3 |
| Bioagência | Brazil | 1.7 | 5.1 | 5.1 |
| CPA Trading S/A | Brazil | 1 | 3.0 | 3.0 |
| CSA Trading | Brazil | 1 | 3.0 | 3.0 |
| Outros |  | 8.6 | 25.5 | 12.7 |
| Subtotal |  | 33.6 | 100 | 62.6 |
| **Sugar** | **Headquarters** |  | **Market share (%)** | **Domestic share (%)** |
| Biosev (Louis Dreyfus) | France/The Netherlands |  | 23 | 0 |
| Alvean (Copersucar/Cargill) | Brazil/USA |  | 22 | 11 |
| Wilmar | Singapore |  | 20.7 | 0 |
| Sucden | France |  | 10 | 0 |
| Outras |  |  | 24.3 | 12.1 |
| Subtotal |  |  | 100 | 23.1 |
| Total (average) |  |  |  | 42.9 |

Source: Novacana, 2022 [26].

Copersucar markets ethanol directly or through Eco-Energy for the case of the the United States market (Copersucar controls 100% of Eco-Energy). Sugar is sold through Alvean, a 50:50 joint venture formed by Copersucar and Cargill. In addition to marketing the production of its partner plants on an exclusive basis, Copersucar markets the sugar and ethanol production of approximately 50 non-partner production units on a non-exclusive basis. In total, Copersucar S.A. trades 2.4 million tons of sugar and 4.8 billion litres of ethanol per year.

The leader in the ethanol segment is the multinational Raízen, with 16.5 billion litres sold annually. The Brazilian sugar market is controlled by four large groups (Biosev, Alvean, Wilmar and Sucden) which together hold around 75% of the traded volume. For the first time in recent years, in 2019, Biosev (controlled by Louis Dreyfus) took the market lead followed by Alvean, Wilmar and Sucden. In addition to trading sugar, Louis Dreyfus controls production at the Santa Elisa, Jardest, Vale do Rosário, Morro Agudo and Continental mills. Singapore-based agribusiness giant Wilmar International increased its stake in 2018 by acquiring sugar trading operations in Brazil from Bunge.

*3.7. Total*

The market shares of Brazilian groups in the main segments assessed are: plant breeding with 93.2% (Table 2); fertilizers with 20.3% (Table 3); machinery with 33.3% (Table 4); mills with 67.4% (Table 5); farming with 75.5 (Table 6); and trading with 42.9% (Table 7). In proportional terms, while foreing companies hold 44.6% of the Brazilian sugar and ethanol supply chain, Brazilian groups hold 55.4% of the entire business (Table 8).

**Table 8.** Proportional participation of domestic groups in the main segments of the sugar and ethanol supply chain in 2021.

| Segment | Market Share (%) | | Domestic Share (%) |
| --- | --- | --- | --- |
| | **National** | **Multinational** | **Proportional to the Total Market** |
| Varieties | 93.2 | 6.8 | 15.5 |
| Fertilizers | 20.3 | 79.7 | 3.4 |
| Machinery | 33.3 | 66.7 | 5.5 |
| Mills (plants) | 67.4 | 32.6 | 11.2 |
| Production | 75.5 | 24.5 | 12.6 |
| Trading | 42.9 | 57.1 | 7.2 |
| Total | | | 55.4 |

Source: Tables 2–7 of this study.

The market share of domestic groups is greater in the segments of plant breeding (with 15.5% of the total domestic proportional participation); farming (with 12.6%) and mills (with 11.2%) (Table 7). In contrast, Brazilian participation is proportionally smaller in segments such as fertilizers (with 3.4% of the total domestic participation), machinery (with 5.5%) and trading (with 7.2%) (Figure 2).

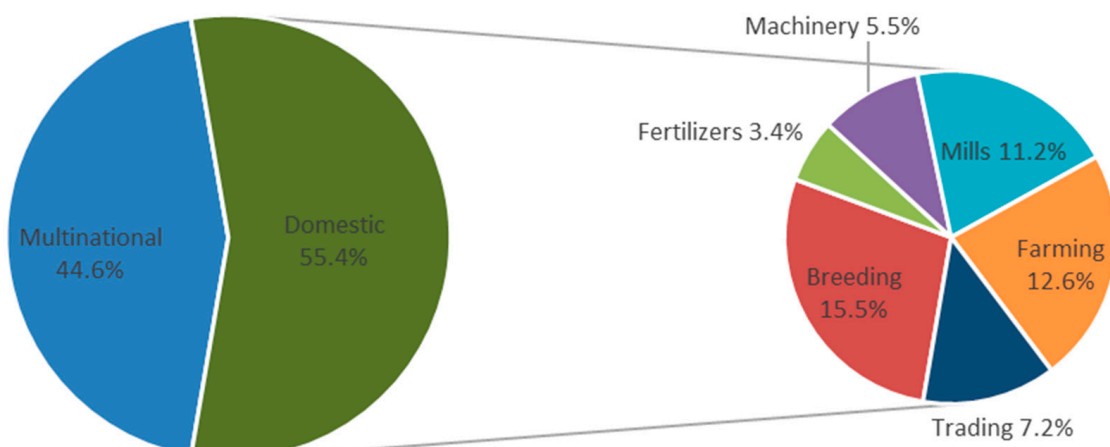

**Figure 2.** Participation of domestic groups and foreign multinationals in the sugar and ethanol supply chain in Brazil in 2021; Source: Table 8 of this study.

## 4. Discussion

This study aims to explore how domestic entrepreneurs can benefit from the thriving global agribusiness by establishing themselves in agro-industrial segments that can best remunerate capital and labour. It is by investing in the agro-industrial sectors, and going beyond the current focus on the primary production of commodities, that developing nations will benefit from agribusiness expansion to build country capabilities and resilience [20].

Based on a detailed assessment of the companies operating in the sugar and ethanol supply chain in Brazil, this study revealed that the current neoliberal economic approach in Brazil resulted in a business in which multinational companies hold 44.6% of the entire supply chain. The control by multinational groups is greater in segments that are intensive in capital and technology protected by patents, such as fertilizers and machinery respectively.

All agro-industrial segments of the sugar and ethanol supply chain established in Brazil have foreign investments made by multinational corporations such as Yara in the manufacture of fertilizers; John Deere, CNH and AGCO in agricultural machinery; Raízen, BP Bunge Bioenergia and Tereos in the mills and Louis Dreyfus in trade. Other important agribusiness supply chains in Brazil, such as soybeans and coffee also have massive foreign multinational investments [13,21].

With economic liberalization, the foreign investments made in Brazil helped to boost agribusiness [11], but it also resulted in a loss of market share by domestic groups due to increasing market concentration by multinationals in some segments [27]. The potential dependency on foreign investments leads to the risk of neocolonialism and the decline of national sovereignty [13].

However, the recent expansion of agribusiness in Brazil also creates areas of opportunity that can benefit domestic entrepreneurs. One of the most well-known examples in Brazil is the expansion of the São Martinho sugarcane group, which is among the largest sugar-energy groups in the world. There are also examples of comprehensive commercial strategies, such as the creation of Copersucar S/A, which is one of the largest global producers and exporters of ethanol and sugar.

This study revealed that by the year 2021, 55.4% of the sugar and ethanol supply chain was held by Brazilian companies, with greater domestic market share in segments such as industrial plants/mills (with 67.4% domestic market share) and farm production (with 75.5% domestic market share). Dynamic markets offer opportunities for developing countries to evolve from their current situation of associated dependency to a new development paradigm with greater market share by domestic companies [10].

The sugar-energy supply in Brazil supply chain also reveals the importance of industrial development policies. Although there has been a reduction in subsidies and state protectionism since the 1990s [19], the supply chain still has government support in the

development of sugarcane varieties (by Ridesa) and in the mandatory blending of anhydrous ethanol in gasoline [16]. Thus, foreign investments from mergers and acquisitions potentially boosted the sector by attracting capital to the country [32], but domestic groups managed to maintain and grow participation in different segments of the supply chain, such as in the development of new sugarcane varieties (with 93.2% domestic market share in sugar cane plant breeding) and trading (with 42.9% domestic market share).

The future of Brazilian agribusiness depends on growing the participation of domestic companies in agro-industrial segments, avoiding the current simplified strategy of farming expansion into new agricultural frontiers with high social and environmental costs [13]. By investing in agro-industrial sectors, developing nations can create country capabilities and resilience [20]. For example, flex-fuel vehicles that run on both ethanol and gasoline benefit consumers in Brazil by proving resilience to gasoline market shocks.

Experiences with new developmentalism in Brazil show the potential of industrial policies that coexist with neoliberal macroeconomic policies [33]. Foreign investments can promote dynamic economic sectors in developing countries, as promoted by the associated dependent development paradigm [7]. Governments can promote local entrepreneurs to participate in these dynamic sectors, producing goods that better remunerate capital and labour, as recommended by the new development paradigm [8]. By having a strategic development approach to engage with global markets, developing countries can explore opportunities to invest in agro-industrial sectors.

## 5. Conclusions

This study revealed that foreign multinational companies control 44.6% of the entire sugar and ethanol supply chain in Brazil, with greater participation in capital and technology-intensive segments such as fertilizers and machinery. However, the expansion of agribusiness also created opportunities for Brazilian groups. Domestic companies hold 55.4% of the entire sugar and ethanol supply chain, with greater participation in segments such as mills (a sector with 67.4% domestic market share) and farming (a sector with 75.5% domestic market share).

The expansion of domestic companies in the agro-industrial sectors can be promoted by strategic policies. The development of the sugar and ethanol supply chain in Brazil has historically benefited from agro-industrial and market policies, which were fundamental to support domestic groups in key segments, such as the breeding of new varieties of sugarcane (a sector with 93.2% domestic market share) and trading (a sector with 42.9% domestic market share). Investments made in science and technology such as in plant breeding tend to support the development of the supply chain as a whole, not only specific companies.

Opportunities created by dynamic economic sectors such as agribusiness can be used by domestic groups to grow their participation in agro-industrial sectors. By investing in agro-industrial capabilities, Brazil can evolve from the current situation of associated dependent development to a new developmentalism-inspired paradigm that seeks to expand domestic participation in sophisticated agro-industrial economic sectors that better remunerate labour and capital and improve country capabilities and resilience. Otherwise, the country runs the risk of remaining in a situation of external dependence with low participation in agribusiness segments that innovate and grow the most.

Developing countries can benefit from agribusiness expansion to build country capabilities, although the impacts are sometimes limited to some sectors where domestic companies can compete with foreign corporations. Lessons from the Brazilian sugar and ethanol supply chain may help other developing countries to evaluate the possibilities and challenges they may face whenever taking similar development pathways focused on agro-industrial development. If in the Brazilian case domestic companies' market shares were relatively small and concentrated in segments less intensive in capital and technology, countries with smaller economies will likely be even more dependent on foreign investments.

**Author Contributions:** Conceptualization, G.d.S.M.; methodology, G.d.S.M.; investigation, G.d.S.M. and R.B.d.C.; data curation, G.d.S.M. and R.B.d.C.; analysis and draft preparation, G.d.S.M. and R.B.d.C.; writing—review and editing, G.d.S.M.; literature review, G.d.S.M. All authors have read and agreed to the published version of the manuscript.

**Funding:** This research received no external funding.

**Institutional Review Board Statement:** Not applicable.

**Informed Consent Statement:** Not applicable.

**Data Availability Statement:** All data used and their source are cited in the manuscript.

**Conflicts of Interest:** The authors declare no conflict of interest.

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
