# Peer review of "Building Agro-Industrial Capabilities in the Sugarcane Supply Chain in Brazil"

_logistics, 2023_

Round 1

Reviewer 1 Report (Previous Reviewer 2)

Now the paper is ready to publish

Author Response

We thank you for your support

Reviewer 2 Report (Previous Reviewer 3)

The article passed by very little improvement from the previous rejected version. Basically, the same issues persist. The main problem regards the reviewe, the theoretical foundation for the analysis and the quality of the refs.
Furthermore, the purpose and the conclusion are misaligned. You target something and retrieved from the findings a conclusion that does not fit with the target.

A review should help. Please strictly avoid refs in other speechs.

Author Response

Dear reviewer, Thank you for your suggestions. We have restructured the introduction, the purpose, the discussion and the conclusion sections to focus on lessons for building agro-industrial capabilities in developing countries. We hope that those changes, including changes made in the title, figure 1 and table 1, and theoretical framework address your concerns.

Reviewer 3 Report (New Reviewer)

Although the study has a great idea, the format, approach, and conclusions did not persuade me. Please review the following comments I have about the paper's content:
1. The research questions, aims, methods, and key findings of the study should be succinctly stated in the abstract. It should give a concise summary of the paper's contribution to the area as well as any research constraints.
2. The review of the literature ought to be thorough and integrated into the discussion. Additionally, it must to point out gaps in the literature and offer ideas for future study directions.

3. The methodology section should provide a thorough explanation of the study's research strategy, data collection procedures, and analytical methods. Along with defending the methods chosen, it ought to go over the approach's advantages and disadvantages.

4. The data analysis needs to be thorough and open. Using the relevant tables, graphs, and statistical analyses, it should show the data and outcomes in a straightforward manner. It should also give thorough justifications for the conclusions and their importance.

5. The study's potential shortcomings and limitations should be acknowledged and addressed in the publication. Any biases, uncertainties, or restrictions in the data, techniques, or analysis should be discussed, along with recommendations for how to deal with them.

6. The paper should be properly arranged and constructed. The introduction, literature review, methods, findings, discussion, and conclusion sections should be very clear. In order to direct the reader, it should also include distinct headings and subheadings.

Extensive editing of English language required

Author Response

Although the study has a great idea, the format, approach, and conclusions did not persuade me. Please review the following comments I have about the paper's content:
1. The research questions, aims, methods, and key findings of the study should be succinctly stated in the abstract. It should give a concise summary of the paper's contribution to the area as well as any research constraints.

Dear reviewer, thank you for your suggestions. We rewrote the abstract to address your suggestions.

  1. The review of the literature ought to be thorough and integrated into the discussion. Additionally, it must to point out gaps in the literature and offer ideas for future study direction

The discussion section has been revised to address that.

3. The methodology section should provide a thorough explanation of the study's research strategy, data collection procedures, and analytical methods. Along with defending the methods chosen, it ought to go over the approach's advantages and disadvantages.

We added table 1 in the methods section to address your suggestion.

  1. The data analysis needs to be thorough and open. Using the relevant tables, graphs, and statistical analyses, it should show the data and outcomes in a straightforward manner. It should also give thorough justifications for the conclusions and their importance.

We added table 1 in the methods section to address your suggestion

  1. The study's potential shortcomings and limitations should be acknowledged and addressed in the publication. Any biases, uncertainties, or restrictions in the data, techniques, or analysis should be discussed, along with recommendations for how to deal with them.

Added in the methods section as follows: The shareholding structure of publicly traded companies is protected by law in Brazil; as a result, multinational companies are not obliged to make this information publicly available. Thus, it was often challenging to determine whether a company was controlled by a Brazilian or multinational company because these multinationals tend not to share this information. A further difficulty arose from the fact that multinational corporations tend to sell their products with the brand names of the Brazilian companies that they acquired

  1. The paper should be properly arranged and constructed. The introduction, literature review, methods, findings, discussion, and conclusion sections should be very clear. In order to direct the reader, it should also include distinct headings and subheadings.

Besides addressing this suggestion we included figure 1 to clearly present the supply chain and the main findings of the study.

Round 2

Reviewer 2 Report (Previous Reviewer 3)

The article improves a little but two shortcomings persist. The purpose is not clear and not entirely justified. Please clarify more what is the purpose of the study and why it is important. The second is the presence of refs in Portuguese. International publication requires only refs in English, otherwise is useless for the audience. Replace all non-English refs by English ones.

Author Response

Dear Reviewer, thank you for your comments. We further edited the abstract and the general objective of the study to clarify its purpose (at the bottom of the introduction section). We also added a new paragraph at the bottom of the conclusion section with the main lessons learned. As for the references, unfortunately, we have no means to address the suggestion since most of the references in Portuguese refer to the source of data used in this study, which was published in Portuguese. As an attempted solution, we translated the references to English for the cases publications were also made available in English. Some papers published in Portuguese were also cited since they were published by scientific journals.

Reviewer 3 Report (New Reviewer)

The revision is made very carefully. Lots of change is done as per suggestion. I strongly recommended it for publication. 

Extensive editing of English language required

Author Response

Dear Reviewer, thank you for your comments and endorsement. We conducted English language editing throughout the manuscript as recommended.

Round 3

Reviewer 2 Report (Previous Reviewer 3)

Please translate titles in Portuguese and inform that the ref is in Portuguese after the pages (in Portuguese)

This manuscript is a resubmission of an earlier submission. The following is a list of the peer review reports and author responses from that submission.

Round 1

Reviewer 1 Report

this study does not have any contribution and methodology to be published as a research study. My decision is rejection. 

Needs to be improved significantly.  

Reviewer 2 Report

Typically, in the Introduction section, you would establish/state, in
that order: (1) Some background to the subject field; (2) The gap in the literature that the paper is addressing, (3) The research questions (which
should be aligned to the gap in the literature), and (4) The novelty and contribution of the paper, which needs to be explicitly discussed and stated. At the moment, the authors have not been discussed and established at all.

Literature reviews are poorly conceptualized and written.

The conclusion is well written, however,  the key differences or new information are not being highlighted. The research problem or research should be
emphasized.

Please cite more scholarly work to support your problem statement.

Reviewer 3 Report

The article handles a very interesting and current problem, the Brazilian sugarcane supply chain. Even if the theme is attractive, the article has minor flaws that prevent publication in the current stage, but are easy to amend:

1.       The abstract is fine but lacks the more important element, the purpose (explicitly announced in the first line). Proceed with motivation, background, methodology, results (explicitly mentioned), and implications;

2.       Please avoid lumped references;

3.       In the end of the Introduction, please state the purpose of the article (one and only one purpose). Also declare what is the methodology and the novelty (why it should be read bu the audience);

4.       The research questions should derive from articulated review that demonstrates the existence of research gaps, there is, the questions were not yet approached by previous studies. A brief review under keywords in a database may help;

5.       The next section should entail a brief review with a figure to explain the main tiers of the sugarcane SC. Furthermore, you employ refs (only 18) in a language other than English (I suppose Portuguese), which turn it inaccessible to a large part of the audience. I suggest replace them by updated articles (> 2017) in English. Also, it should inform step-by-step the methodology you employed;

6.       Strongly avoid footnotes in scientific articles. If the content is relevant, it must migrate to the body of the text, perhaps a specific table;

7.       The last two sections could be merged under a new name, Final Remarks.

8.       I believe you have too few refs. Try your best to improve your refs list in order to ensure reliability to your study. Focus only in peer-reviewed studies not before 2017.

Best regards